# Oral Microbiota: A Major Player in the Diagnosis of Systemic Diseases

**DOI:** 10.3390/diagnostics11081376

**Published:** 2021-07-30

**Authors:** Charlotte Thomas, Matthieu Minty, Alexia Vinel, Thibault Canceill, Pascale Loubières, Remy Burcelin, Myriam Kaddech, Vincent Blasco-Baque, Sara Laurencin-Dalicieux

**Affiliations:** 1INSERM UMR 1297 Inserm, Institut des Maladies Métaboliques et Cardiovasculaires (I2MC), Avenue Jean Poulhès 1, CEDEX 4, 31432 Toulouse, France; alexia.vinel@inserm.fr (A.V.); pascale.loubieres@inserm.fr (P.L.); remy.burcelin@inserm.fr (R.B.); vincent.blasco@inserm.fr (V.B.-B.); 2Faculté de Chirurgie Dentaire, Université Paul Sabatier III (UPS), 118 Route de Narbonne, CEDEX 9, 31062 Toulouse, France; thibault.canceill@sfr.fr (T.C.); myriamkaddech@hotmail.com (M.K.); laurencin.s@chu-toulouse.fr (S.L.-D.); 3Service d’Odontologie Rangueil, CHU de Toulouse, 3 Chemin des Maraîchers, CEDEX 9, 31062 Toulouse, France; 4UMR CNRS 5085, Centre Interuniversitaire de Recherche et d’Ingénierie des Matériaux (CIRIMAT), Université Paul Sabatier, 35 Chemin des Maraichers, CEDEX 9, 31062 Toulouse, France; 5INSERM UMR 1295, Centre d’Epidémiologie et de Recherche en Santé des Populations de Toulouse (CERPOP), Epidémiologie et Analyse en Santé Publique, Risques, Maladies Chroniques et Handicaps, 37 Allées Jules Guesdes, 31000 Toulouse, France

**Keywords:** oral microbiota, systemic disease, dysbiosis, periodontitis, endotoxemia, inflammation

## Abstract

The oral cavity is host to a complex and diverse microbiota community which plays an important role in health and disease. Major oral infections, i.e., caries and periodontal diseases, are both responsible for and induced by oral microbiota dysbiosis. This dysbiosis is known to have an impact on other chronic systemic diseases, whether triggering or aggravating them, making the oral microbiota a novel target in diagnosing, following, and treating systemic diseases. In this review, we summarize the major roles that oral microbiota can play in systemic disease development and aggravation and also how novel tools can help investigate this complex ecosystem. Finally, we describe new therapeutic approaches based on oral bacterial recolonization or host modulation therapies. Collaboration in diagnosis and treatment between oral specialists and general health specialists is of key importance in bridging oral and systemic health and disease and improving patients’ wellbeing.

## 1. Introduction

The oral cavity is host to a complex microbial group, the oral microbiota, composed mainly of bacteria but also includes viruses, protozoans, fungi, archaea, phages, and ultra-small bacteria belonging to the candidate phyla radiation group [1,2,3]. Oral microbiota, developed from the first minutes of a child’s life, is composed of more than 500 different bacterial species [4,5]. Its exact composition varies greatly from one individual to another as it depends on several factors, such as age, diet, and lifestyle habits (such as smoking and physical activity) [6]. Present in a physiological state, these bacteria cohabit with the human body in a state of symbiosis. There are two equilibriums; one between the proliferation of the different species and the immune capacities of the host and the other, between the bacterial species themselves. In the event of dysbiosis, i.e., a breakdown in this homeostasis, one or more types of microorganisms proliferate and take over—at least temporarily—the immune system [7]. Various oral pathologies can then occur and will depend on the type of bacterial imbalance [8]. However, the oral microbiota does not only have a local function. We now know that our mouth, through its microbiota, communicates intensely with our whole body [3]. Indeed, oral dysbiosis causes a localized inflammatory state in the oral sphere that can contribute to the maintenance of chronic low-grade systemic inflammation [9]. This can lead to the development or aggravation of systemic pathologies, sometimes with a significant degree of morbidity. The identification of bacteria responsible for these relationships between the oral cavity and the rest of the body is therefore of major interest. The oral microbiota thus becomes a marker of our general health via its systemic activity. The aim of this review is to describe the role of oral microbiota and its bacterial effectors on our general health. The oral cavity and its microbiota play an important role in the diagnosis of an individual’s health status and may also give rise to new preventive and therapeutic approaches [10].

## 2. Oral Microbiota

The oral cavity is the second-largest microbiota reservoir in the body and has been the subject of intense study for several years [11]. The accessibility of the site has made the oral microbiota one of the best-known bacterial communities in the human body, with more than 500 different species identified in adults. Even if the maternal microbiota already plays an important role in the neurodevelopment of the fetus before birth [12,13], the acquisition of the microbiota begins in the first minutes of life through direct contact with the maternal microbiota (skin, vagina, and mouth) [14]. The child’s environment is also essential to the construction of its microbiota which is not solely acquired through heredity [4]. 

The different ecological niches that make up the oral microbiota consist mainly of hard and soft tissues [15,16]. Saliva has a major influence on the oral microbiota and participates in maintaining oral homeostasis [17]. It is essential in mechanical cleansing, has a buffering power to maintain a physiological pH and antimicrobial properties through its various non-specific immune components (immunoglobulin A, lysozyme, lactoferrin, peroxidase system, histidine-rich proteins, etc.). The composition of saliva plays a determining role on the microbiota through a cascade of mechanisms linked to the molecules in suspension, which condition the composition of the biofilm. The latter allows the attachment of bacteria to oral surfaces. Salivary components such as glycoproteins are a source of nutrition to ensure the growth of bacteria present in the mouth [17,18]. All these bacteria act synergistically, at very low concentrations, allowing a complex balance between the surrounding microbiota and the host’s oral cavity. An insufficient salivary flow can easily lead to dysbiosis.

Bacteria are present in the mouth in two forms: either in a planktonic state, as in saliva, or in a biofilm as in the dental plaque. Bacteria present in planktonic form circulate freely in the oral environment, whereas dental plaque is the result of the heterogeneous accumulation of aerobic and anaerobic bacteria that form an adherent deposit on the surface of teeth and oral mucosa [19]. Dental surfaces are covered with an organic film called the Acquired Exogenous Pellicle (AEP), which protects teeth from mechanical and acidic aggressions but can also condition the tooth surface to promote further attachment of bacteria as part of the biofilm developmental process. Indeed, the planktonic bacteria present in saliva can adhere to the AEP in a reversible way by electrostatic and Van Der Waals forces or in an irreversible way by ionic or covalent bonds. The first step in the creation of a biofilm is the irreversible adhesion of pioneer bacteria such as *Streptococcus gordonii*, *Streptococcus oralis*, or *Streptococcus mitis* to the AEP, which then proliferates. Secondly, these pioneer bacteria allow the aggregation of new bacteria called late or secondary colonizers [20]. As the biofilm matures, an equilibrium is created between each bacteria present. Microbial dysbiosis can occur due to local and systemic diseases resulting in a gradual shift toward bacteria with specific profiles.

Oral microbiota dysbiosis is at the origin of two major oral pathologies: caries and periodontitis. However, these two pathologies have different profiles [7]. Caries are responsible for the destruction of the tooth’s hard tissues. They are caused by salivary dysbiosis (reduction of the bacterial diversity) resulting from a disturbed supra-gingival biofilm associated with an excess of sugar consumption and/or poor oral hygiene (factors of dysbiosis). This imbalance leads to a modification in the biofilm composition characterized by an increase in Gram-positive bacteria and acidification of the medium [21,22]. The main pathogens increased are *Streptococcus mutans*, *Actinomyces*, and *Lactobacillus*. *Bifidobacterium* spp., *Scardovia* spp., and *Candida albicans* are also found in greater quantities [23,24,25]. This increased acidification is accompanied by a loss of diversity and a reduction in the level and metabolic activity of beneficial bacteria, which prefer to grow at neutral pH. Periodontitis is an oral disease resulting from periodontal dysbiosis associated with a deleterious immuno-inflammatory reaction of the host, leading to the progressive and irreversible destruction of the dental attachment system (the periodontium) and ultimately causing tooth loss. In contrast to the rest of the oral cavity, the microorganisms that develop in the sulcus and adhere to the root surface evolve in an environment that is less rich in oxygen and more protected from shear forces than those found at the supra-gingival level. Periodontal dysbiosis is characterized by an increase in the proportion of Gram-negative anaerobic bacteria historically described in 1979 as belonging to Socranski’s “red complex”: *Treponema denticola* (*Td*), *Porphyromonas gingivalis* (*Pg*), *Tannerella forsythia* (*Tf*), and *Fusobacterium nucleatum* (*Fn*) [26]. *Prevotella intermedia* (*Pi*), *Dialister* spp., and *Selenomonas* spp. are also found in abundance in periodontitis, and the high number of spirochetes seems to be associated with the severity of periodontal destruction [27,28,29,30]. However, metagenomic sequencing has made it possible to highlight new concepts concerning periodontal dysbiosis associated with periodontitis. This dysbiosis is the result of a qualitative and quantitative modification of a polymicrobial community, including commensal and pathogenic bacteria [31]. At the periodontal level, dysbiosis comes more from a change in dominant species than from de novo bacterial colonization. Contrary to what is found at the intestinal level, periodontitis is associated with an increase in the diversity of its microbiota [32,33,34]. This greater diversity is thought to be the result of the additional nutrient supply from the destruction of surrounding tissues and the physical increase in the size of the periodontal pocket lesion [7,28].

## 3. Physiopathologic Mechanisms of Dysbiotic Oral Microbiota and Systemic Inflammation during General Diseases

Recent data from the scientific literature suggest that there is a link between the development and/or aggravation of certain systemic pathologies and the occurrence of an imbalance in the oral flora [7]. These diseases have in common a pathophysiological mechanism based on the maintenance of a systemic inflammatory state (low-grade inflammation) whose resolution or stabilization is not possible in the presence of dysbiosis in the oral microbiota. Some authors also suggest that dysbiotic oral microbiota-causing microorganisms could circulate systemically or release part of their constituents into the bloodstream and thus cause complications away from their original site of proliferation. Bacterial meta factors are all the bacterial parts that have an activity of either virulence or activation of the immune system during physiology and physiopathology (for example, lipopolysaccharide (LPS), Flagellin, and teichoic acid). One of the main mediators of this immuno-inflammatory disorder is LPS, which is an endotoxin and a major virulence factor of Gram-negative bacteria. LPS is an example since it is likely to migrate to different organs (liver, muscles, heart, etc.) and trigger inflammatory reactions there [35]. Indeed, bacteria or their components, such as LPS present on the outer membrane of Gram-negative bacteria, are a continuous source of inflammation and infection which cross the epithelial barrier, arrive in the bloodstream, and reach various organs of the body. The main cause of this bacterial translocation is oral microbiota dysbiosis. That is to say, a loss of the balance of the salivary bacterial composition due to an excess of Gram-negative or Gram-positive bacteria or virulence factors (LPS, Flagellin, teichoic acid). Oral bacteria or meta factors activate innate immune defenses via Pathogen-Associated Molecular Patterns (PAMP) recognized by Pattern Recognition Receptors (PRR) of innate immune cells. PAMPs are represented by different surface molecules such as teichoic acid (Gram-positive specific) or LPS (Gram-negative specific). PRRs include a group of receptors called Toll-Like Receptor (TLR) [36]. The interaction between TLRs and PAMPs leads to the activation of the MyD88 (Myeloid Differentiation protein 88) signaling pathway and then that of the transcription factor NF-kappa B (NF-kB), inducing a generalized inflammatory response with the synthesis of a variety of pro-inflammatory mediators, such as Tumor Necrosis Factor α (TNF-α) and interleukins.

Figure 1 represents the different possible pathways (by bacterial translocation in the blood) responsible for an increase in pro-inflammatory cytokines in systemic diseases resulting from oral microbiota dysbiosis. The transduction of an inflammatory signal to the nucleus of cells requires a succession of phosphorylation chain reactions catalyzed by protein kinases. Activation of the inflammasome is carried out according to several pathways, those of MAPK (Mitogen-Activated Protein Kinases), IKKs (Inhibitor of Kappa B Kinases), and those of JAK/STAT (Janus Kinase and Signal Transducers and Activators of Transcription). The dissemination throughout the body of these oral bacteria or their components (LPS) is at the origin of a modification of the bacterial balance and, therefore, the microbiota of other organs. Several studies have shown other ways of translocation of these oral bacteria by different processes such as ingestion (intestinal microbiota), inhalation (pulmonary microbiota), or sexual intercourse (vaginal microbiota). This shows the link between the oral microbiota and the different microbiota of the organism at the origin of various systemic pathologies described below.

In fact, oral microbiota and its balance play a major role in an individual’s general homeostasis. Any disruption leads to an increase in certain bacterial species, especially Gram-negative ones, associated with the massive production of pro-inflammatory cytokines, which causes or maintains chronic low-grade inflammation. In this review, we discuss the role that dysbiotic oral microbiota may have on different systemic pathologies: metabolic diseases, cardiovascular diseases, respiratory diseases, rheumatoid arthritis, adverse pregnancy outcomes, inflammatory bowel diseases, Alzheimer’s disease, autism spectrum disorders, and oral mucosal diseases (Figure 2). 

### 3.1. Metabolic Diseases

Many pathologies are likely to be influenced by this low-grade inflammation. Most of them are represented by the so-called metabolic diseases, which affect the cellular functions of energy production and storage, or even in a broader sense by the “metabolic syndrome” (MetS). This corresponds to a combination of at least three manifestations among obesity, hypertriglyceridemia, high blood pressure, hyperglycemia, and reduced levels of high-density lipoproteins (HDL). Many studies have investigated the role of oral microbiota in these metabolic diseases [37]. At the epidemiological level, there is a positive association between periodontal disease and metabolic pathologies: patients with MetS have a higher risk of developing periodontitis; conversely, patients with periodontitis have a higher risk of suffering from MetS [37,38,39]. However, the molecular and microbiological mechanisms responsible for these clinically identified associations are still poorly understood [40,41]. The local inflammation associated with oral dysbiosis contributes to maintaining/aggravating a significant metabolic, inflammatory state throughout the body [37] and can cause insulin resistance, inflammation, vascular, and metabolic disorders. Conversely, the lack of recovery from metabolic symptoms maintains the pathological state at the oral level [42].

Diabetes is the metabolic disease whose relationship with oral microbiota has been the most extensively explored. Diabetes is a chronic metabolic disorder defined by fasting hyperglycemia and is due to a deficit in insulin secretion by pancreatic β-cells of autoimmune origin for type 1 diabetes (T1D) or decompensated insulin resistance for type 2 diabetes (T2D) [43,44]. In 2019, the number of people diagnosed with diabetes worldwide was 463 million, or 6% of the world’s population. Interestingly, 80% of T2D could be avoided with a healthier diet and regular exercise [45]. However, in recent years, despite the management of these risk factors, the number of diabetics continues to rise, suggesting that there are other risk factors for diabetes, such as oral and gut dysbiosis [46,47].

The bidirectional link between diabetes and periodontitis has been demonstrated with inflammation as a common mediator [48], making periodontitis the sixth complication of T2D [49] and also suggesting that the “motor” of this pathology could be chronic low-grade inflammation [50], which is itself the result of oral microbiota dysbiosis. Differences in the oral and periodontal microbiota were found in diabetic subjects compared to healthy subjects [44,47,51]. A significant increase in the genera *Aggregatibacter*, *Neisseria*, *Gemella*, *Eikenella*, *Selenomonas*, *Actinomyces*, *Capnocytophaga*, *Fusobacterium*, *Veillonella*, and *Streptococcus* was observed in people with diabetes [47,52]. In addition, the involvement of oral pathogens such as *Pg* has been demonstrated in insulin resistance [53]. Indeed, high levels of TNF-α and Interleukin-6 (IL-6) (produced by periodontal macrophages in the presence of *Pg*) increase the permeability of the epithelial barriers of the oral cavity [54] and thus favors the passage of Gram-negative bacteria and their virulence factors such as LPS into the bloodstream. In addition to the inflammatory state this causes in the organs, LPS are also responsible for inhibiting the transduction of insulin receptor-initiated signaling pathways and thus the development of insulin resistance [55]. Furthermore, periodontal treatment could reduce glycated hemoglobin (HbA1c) levels by up to 0.4% in T2D patients [56]. Also, oral microbiota can influence the progression of diabetes [57], and some oral hygiene measures, such as overuse of mouthwashes, can even have a detrimental effect on the progression of diabetes [58]. Nonetheless, many mechanisms on this two-dimensional interaction still remain to be elucidated.

The link between gut microbiota and obesity is already established, and numerous studies now focus on the link with the oral microbiota [59,60,61,62]. Literature has described that, as with the intestinal microbiota, there is a modification of the oral microbiota in obese people with an increased Firmicute/Bacteroides ratio [63]. Several authors have found differences in the composition of the oral microbiota between obese and normal-weighted subjects with a particular increase in certain genera and species in obese subjects: *Peptostreptococcus*, *Solobacterium*, *Selenomonas noxia*, *Pg*, *Pi*, and *Tf* [63,64,65,66,67]. Recently we showed that obese people present an increased periodontal risk associated with an increase of *Capnocytophaga* present in the oral microbiota. Moreover, we showed that sex/gender plays a role in the oral microbiota signature of obesity in subjects with periodontitis: obese females were characterized by an increase in the *Streptococcus* genus compared to obese males, where the *Neisseria* genus was increased [68]. Even if the positive association between obesity and periodontitis has been demonstrated in numerous clinical and epidemiological studies, the causal link between oral microbiota and obesity remains to be clarified. Finally, obesity could lead to dyslipidemia and high blood pressure, which in return induce cardiovascular diseases. 

### 3.2. Cardiovascular Diseases

According to the World Health Organization (WHO), cardiovascular diseases (CVD) are the first cause of death worldwide, and deaths from cardiovascular and circulatory diseases are continuously rising, partly due to population growth and aging [69,70]. An estimated 17.9 million deaths per year are attributable to cardiovascular diseases, or 32% of global mortality [71]. There are many risk factors for cardiovascular diseases such as gender, age, smoking, dyslipidemia, hypertension, insulin resistance, and overweight and obesity [71,72,73]. However, despite the management of these factors, the mortality rate continues to rise associated with inter-individual variability in the risk of unexplained cardiovascular mortality and morbidity, which leads to the search for new risk factors. An increasingly studied risk factor is the immuno-inflammatory axis [74], and oral microbiota dysbiosis is a major risk factor in the development of cardiovascular disease [75]. 

One of the main causes of cardiovascular disease is the development of atherosclerosis under the influence of environmental and genetic factors. Atherosclerosis results from the accumulation of low-density lipoproteins (LDL) in the vascular wall, where they undergo oxidation, causing endothelial dysfunction [76]. Endothelial cells then allow the transmigration of circulating inflammatory cells, monocytes, that differentiate into macrophages within the subendothelial space where they phagocyte the oxidized LDL, becoming foamy cells. Then, nearby vascular smooth-muscle cells migrate to the lesion site and synthesize a protein matrix, forming a fibrous cap above the thrombogenic lipidic core. The more the plaque grows, the narrower the vessel lumen becomes, and in case of plaque rupture, the exposed lipidic core activates platelets leading to a thrombus able to block the blood flow in situ or downstream completely. Atherosclerosis is thus the main cause of myocardial infarction, coronary heart disease, stroke, and peripheral artery disease [76].

The first to raise the link between myocardial infarctions and poor oral health were Mattila et al. in 1989 [77]. Since then, numerous studies have concluded a positive association between periodontal diseases and CVD independently of their common risk factors (smoking, age, diabetes) [75,78,79,80,81]. Moreover, periodontal treatment has been shown to have beneficial effects on cardiovascular biochemical parameters such as endothelial function and C-Reactive Protein (CRP) levels [82,83,84,85]. The mechanisms linking periodontitis and CVD are both direct, via bacterial invasion, and indirect, via inflammation. Indeed, several periodontal pathogens have been identified in the atheromatous plaque, such as *Pg*, *Aggregatibacter actinomycetemcomitans* (*Aa*), *Tf*, *Pi*, *Td*, and *Fn*, [86,87,88,89,90,91]. Koren et al. have demonstrated the presence of oral bacteria DNA (*Chryseomonas*, *Veillonella*, and *Streptococcus*, mainly) on samples of atheromatous plaques and were able to establish a correlation between the measured quantity of some of these DNAs and the significant quantity of the corresponding bacteria in the oral cavity of patients [92]. *Pg* DNA has also been detected in the cardiac valves of patients with CVD and deep periodontal pockets and even in healthy vascular tissues. *Pg* is the most abundant specie in patients with atherosclerosis after bypass surgery [93,94]. In vivo studies showed that the administration of periodontal pathogens to different mice models has notable vascular effects: increased vascular reactivity, smooth muscle cells proliferation and aortic plaque development with *Pg* [95,96,97], arterial invasion and reduced nitric oxide levels with *Td* [98], higher levels of inflammatory cytokines such as IL-6 and CRP, LDL and atherosclerotic lesion progression with *Fn* [99]. Moreover, polymicrobial infection with *Pg*, *Td*, *Tf*, and *Fn* is associated with accelerated atherosclerosis in vivo [100,101]. Recent studies showed that *Pg* is able to increase vascular permeability and cause vascular damages via, respectively, its outer membrane vesicles and surface-expressed gingipains in vitro and in vivo in a zebrafish model [102]. Also, Pussinen et al. found that coronary heart disease is more common in people who have anti-*Pg* antibodies compared to those who do not, suggesting that periodontal infection or the host’s response to anti-*Pg* infection could play a role in the pathogenicity of coronary heart disease [103,104].

The second mechanism linking periodontitis and CVD involves the increased inflammatory cytokines systemic levels occurring in both pathologies. The high levels of Interleukin-1β (IL-1β), IL-6, Interleukin-8 (IL-8), TNFα, and monocyte chemoattractant protein-1 (MCP-1) associated with periodontitis can induce a rapid hepatic synthesis and secretion of plasmatic vascular proteins such as CRP and fibrinogen [105,106]. LPS produced by periodontal pathogens and released in the blood flow triggers an immunological response which, when associated with CRP and fibrinogen, may initiate atherosclerosis by acting on endothelial cells, lipid metabolism modulators, and enhancing oxidative stress [107,108]. LPS is transported and eliminated from the bloodstream by lipoproteins. It is associated with healthy people with HDL present, which helps in its neutralization in case of infection. When HDL levels decrease sharply, it mainly associates with VLDL (Very Low-Density Lipoproteins) [75,109]. LPS associated with VLDL crosses the barrier of endothelial cells to the intima, causing an alteration of lipid homeostasis. VLDL also potentiates the expression of TNF-α and MCP-1 by macrophages resulting in their activation [109]. This systemic response leads to an increase in the production of cytokines associated with an alteration in lipid metabolism. Patients with a lipoprotein profile predominantly consisting of small, dense LDL are three to seven times more likely to have a cardiovascular event than people with “normal” lipoprotein profiles. Studies have shown that periodontal infections could be correlated with this type of lipid profile [110].

In addition, dyslipidemia is more common in patients with periodontitis. It manifests itself by an increase in the levels of LDL and plasma triglycerides with, conversely, a decrease in HDL. A study has shown that patients with moderate periodontitis do not show quantitative but qualitative changes in LDL because there is an increase in VLDL as previously described [111]. In addition, the elevated levels of cholesterol and LDL noted in the bloodstream of patients with atherosclerosis have been positively correlated with the proliferation of *Fusobacterium* in the microbiota of their oral cavity [92].

### 3.3. Chronic Obstructive Pulmonary Disease

Although lungs have long been regarded as sterile, bacterial colonization begins soon after birth and is a dynamic process influenced by environmental and genetic factors. Bacteria from the upper respiratory tract and the environment reach the lower respiratory tract through breathing, mucociliary clearance, and microaspiration [112]. In healthy subjects, there are important similarities between oral and lung microbiota [113,114], and a microbiome dominated by *Streptococcus*, *Prevotella*, *Veillonella*, *Pseudomonas*, *Haemophilus*, and *Fusobacterium* has been described in the respiratory tract [115]. While the mechanisms have yet to be understood, lung microbiota changes in chronic respiratory diseases are increasingly studied. 

Chronic Obstructive Pulmonary Disease (COPD) is characterized by dyspnea, chronic cough, and sputum, resulting from chronic inflammation within the respiratory airways [116]. Lung microbiota evolves according to disease severity [117] and exacerbations, characterized by alveolar rupture, increased mucus production, and disease aggravation, which are mainly caused by microorganisms promoting inflammatory mediators’ production by inflammatory cells [118,119]. Studies showed that *Haemophilus influenzae*, *Streptococcus pneumoniae* (*Sp*), *Pseudomonas aeruginosa* (*Pa*), and *Moraxella catarrhalis* are the main microorganisms involved in acute exacerbations of COPD (AECOPD) [118,120,121] and interestingly, a study showed the presence of *Sp* and *Pa* in subgingival plaque and *Pg* and *Td* in the tracheal aspirates of patients suffering from acute exacerbation of COPD [122]. Several clinical studies showed that periodontitis is associated with an increased risk of developing COPD, independently of conventional risk factors (tobacco, age) [123,124]. Moreover, periodontal treatment is accompanied by a reduction of exacerbations and respiratory parameters [125,126]. Recent studies showed that *Fn* is able to induce proinflammatory cytokines IL-6 and IL-8 production by respiratory epithelial cells in vitro and in vivo [127]. Moreover, *Fn* and *Pa*, which coexist in the respiratory system of patients suffering from AECOPD, can collaborate to sustain bacterial growth in a co-culture model [128]. Simultaneous coinfection of human respiratory epithelial cell lines with *Fn* and *Pa* promotes cell invasion and induces IL-6, IL-8, and TNF-α in vitro [129].

Besides COPD, a recent study showed that periodontitis seems to be associated with increased risk of intensive care unit admission, need for assisted ventilation, and death of patients suffering from SARS-CoV-2 [130]. In vitro experiments suggest that the aspiration of periodontopathogen bacteria like *Fn* by patients with COVID-19 may aggravate lower respiratory tract inflammation [131].

### 3.4. Rheumatoid Arthritis

Rheumatoid arthritis (RA) is an auto-immune disease characterized by chronic inflammation of the joints. At the anatomopathological level, the synovial membrane, when inflamed, forms what is called a “pannus,” which leads to the destruction of the joints. Its incidence is 0.5% to 1% and concerns mostly women between the ages of 40 and 60 [132]. In 50 to 70% of cases, patients produce autoantibodies against citrullinated peptides (ACPAs) and autoantibodies against IgG (RF, for rheumatoid factor) [133]. ACPAs are able to form immune complexes with citrulline-containing antigens that then link the RF, leading to important complement activation [134]. This immune activation is responsible for synovial membrane inflammation along with leucocyte infiltration into the synovial compartment of the joints [133]. This inflammation induces fibroblast activation, enhanced chondrocyte catabolism, and synovial osteoclastogenesis, leading ultimately to articular destruction [135]. Interestingly, circulating ACPAs can be detected long before RA diagnosis [136,137,138].

For several years, the role of periodontitis and its microbiota has been studied in the development of rheumatoid diseases such as RA. Indeed, studies have shown that patients with RA are more likely to develop chronic periodontitis compared to healthy people [139]. Other studies have shown that patients with RA have a significantly higher prevalence of moderate to severe periodontitis (62.5%). In addition, patients with periodontal follow-up (therefore associated with periodontitis) have a higher prevalence of RA (4%) than the general population (1%) [132]. These two inflammatory pathologies develop under the influence of many common risk factors, whether genetic with the polymorphism of certain receptors, but also immune with increased production of interferon-gamma, B lymphocytes, and RANKL (Receptor Activator of Nuclear Factor-Kappa B Ligand) associated with activation of the inflammation system with a TH17 (lymphocyte T Helper cells expressing IL-17) profile [140]. The presence of bacteria in the synovia and an increased level of inflammatory mediators in the blood (CRP) were also common between these two pathologies [141]. Thus, there is a bidirectional link between periodontitis and RA: patients with periodontitis are more likely to suffer from RA, and conversely, patients with RA have a greater risk of periodontitis [138,142,143,144].

Chronic periodontitis is believed to be involved in the initiation and progression of RA mainly by two major biological pathways. First, the passage of bacteria and pro-inflammatory cytokine of periodontal origin in the bloodstream favoring the development of joint inflammation. In fact, many microorganisms are mentioned in these hypotheses, such as Mycoplasmas, viruses (Epstein Barr virus and Cytomegalovirus), and bacteria of the periodontal microbiota. The presence of bacterial DNA and peptidoglycans, constituents of their wall, in the joints of patients with RA has been demonstrated. Among periodontal bacteria, the most regularly detected are *Pi* and *Pg*, as well as *Bacteroides forsythus* [145]. The prescription in these patients of antibiotic treatments against oral anaerobic bacteria leads to an improvement in the clinical signs of RA.

Secondly, the link between periodontitis and RA has been studied through the specific role of *Pg* due to its enzymes, the peptidyl-arginine-deiminases (PAD), which could carry out the citrullination (post-translational modification) of certain peptides [146]. The result is the production of antibodies that recognize the host’s citrullinated peptides, which is indicative of clinical signs of RA. *Pg* is able to citrullinated bacterial and human proteins, leading to the formation of APCAs, and potentially contributing to the initiation of RA [147,148]. The positive assay of these ACPAs makes it possible to predict, with a specificity greater than 95%, the early diagnosis of RA. Studies show that the level of anti-*Pg* antibodies is correlated with the level of ACPAs in patients with RA. The concentration of anti-*Pg* antibodies is increased in RA patients compared to healthy subjects and detectable several years before the appearance of RA symptoms [149]. More recently, a study showed that *Aa* is also able to release hyper-citrullinated proteins and that there is a strong association between the concentration of anti-*Aa* antibodies and the presence of ACPAs and RF in patients with RA [150].

While the effects of periodontal treatment are controversial, with conflicting results among studies, and well-conducted clinical trials are needed to conclude on this point [151,152,153,154] firmly, it seems that anti-inflammatory treatment for RA induces a reduction in gingival inflammation (IL-1β) [155]. Interestingly, an in vivo study on mice showed that ACPAs serum concentration is reflected in the saliva. A human study observed alteration of oral microbiota in the pre-clinical stages of RA, suggesting that oral status could be an early marker of RA development [151,156]. There is, therefore, an interrelation between these two inflammatory pathologies with significant production of pro-inflammatory cytokines [152].

### 3.5. Adverse Pregnancy Outcomes

During pregnancy, physiological, immunological, and hormonal modifications occur in the mother, increasing her susceptibility to infections, including oral and periodontal diseases [153,154]. Also, changes in estrogen and progesterone levels affect the composition of the oral microbiota, which becomes compatible with the development of gingivitis and periodontitis [157,158]. Oral saliva of pregnant women that underwent 16S rRNA gene sequencing shows that the oral microbial diversity is relatively stable during pregnancy. However, compared to non-pregnant women, it makes a pathogenic shift during pregnancy associated with pregnancy gingivitis before reverting to a healthy microbiome during the post-partum period [159]. Also, oral dysbiosis, as described in periodontitis, can increase the risk of adverse pregnancy outcomes (APO), including preterm birth, preeclampsia, gestational diabetes, and low birthweight [160,161]. Furthermore, maternal periodontal conditions seem to be associated with stillbirth and perinatal death [162,163]. Three biological mechanisms supporting this relationship between periodontal diseases and APO have been proposed: one through systemic dissemination of periodontal pathogens that could cross the placenta into the amniotic fluid and fetal circulation, the other through inflammatory mediators such as IL-6, IL-8, and TNF-α produced during periodontal inflammation that could, by entering the systemic circulation, induce an acute inflammatory response affecting the fetus and placenta, and finally, the possibility of oral microbiological transmission to the vaginal microbiome resulting from sexual practice [9,164,165]. Also, the placenta harbors a unique microbiome structured by the history of antenatal infections and shares similarities with the oral microbiome [166]. Recent studies have shown that both placental and oral microbiomes may play a role in periodontitis-associated APO [167]. *Pg* and *Fn* Gram-negative anaerobes commonly found in periodontitis have been shown to be able to translocate to the fetal-placental unit [168,169] and are associated with APO such as preeclampsia, early-onset neonatal sepsis, and stillbirth [167,170].

Finally, concerning the transmission of the maternal microbiome to the child, recent research on families with adopted and biological children shows that the composition of the microbiome seems to be shaped more by the host and contact with the local environment than by genetics and direct transfer of this microbiome through pregnancy outcomes and especially delivery modes (C-section or vaginal delivery) [4]. Babies fed only on breast milk have a lower diversity with a decreased relative abundance of *Veillonella*, *Prevotella*, *Granulicatella*, and *Porphyromonas* than those fed only on formula, but these differences do not persist over time. Also, at two months, the salivary microbiome of infants delivered by C-section is significantly more diverse than those delivered vaginally, but again these differences seem to disappear at 12 months [171]. The salivary microbiome is dynamic during the first two years of life, and age-related factors seem to be the strongest determinants. Colonization by species such as *Candida albicans* is a good example of these age-related changes in exposure [171].

### 3.6. Inflammatory Bowel Diseases

Chronic inflammatory bowel diseases (IBD) mainly include Crohn’s disease and ulcerative colitis. These pathologies are characterized by chronic intestinal inflammation evolving into a variable course of flare-ups and remissions. Although the etiology is still imperfectly known today, the main hypothesis is an inadequate intestinal immune response to bacteria in the commensal flora, probably triggered by environmental factors associated with genetic predispositions. Extra-digestive manifestations at the musculo-articular, cutaneous, oral, and ophthalmological levels are the most frequent and often progress in parallel to digestive disease outbreaks [172,173]. The oral cavity is a preferential site for extra-digestive manifestations: 50% of IBD patients develop oral lesions with a wide variety of clinical forms [174]. They may be specific to intestinal pathology (inaugural, at the same time, or after the appearance of gastrointestinal symptoms), or they may be reactive, i.e., the consequence of drug treatments and nutritional deficiencies due to malabsorption and malnutrition (so-called non-specific lesions) [175,176]. They may also be an early sign of therapeutic escape or disease relapse. People suffering from IBD are characterized by dysbiotic oral and gut microbiota with a decrease in bacterial diversity, a decrease in the abundance of Firmicutes phylum, and an increase in the abundance of Proteobacteria and Bacteroidetes phylum [102,177,178,179]. Recent studies showed that, compared to healthy people, these people also presented an oral dysbiosis and that periodontitis was significantly associated with IBD [180,181]. However, the exact nature of oral dysbiosis remains unclear and differs between studies. Some authors observed an increase in *Eikenella corrodens*, *Prevotella* genus, *Veillonella* genus, and a decrease in *Streptococcus* genus, *Haemophilus* genus, *Neisseria*, and *Gemella* families in IBD patients compared to healthy subjects [182,183]. However, others observed an increase in *Campylobacter* genus, Streptoccocaceace, Enterobacteriaceae, and Veillonellaceae families and a decrease in Porphyromonadaceae and Neisseriaceae families [184,185,186]. Furthermore, the oral microbiota of Crohn’s disease patients was associated with a decrease in *Neisseria*, *Haemophilus*, *Fusobacterium*, and *Porphyromonas* genera compared to patients in remission [187]. In addition, an increase in the number of oral bacteria in the gut microbiota has been observed in IBD patients: *Aggregatibacter*, *Campylobacter*, *Enterobacteria*, *Fusobacterium*, *Gemella*, *Neisseria*, *Veillonella*, *Peptostreptococcus*, and *Streptococcus* [188,189,190]. Therefore, it can be assumed that the oral microbiota, capable of colonizing the gastrointestinal tract, may act as a pathogen reservoir and thus play a role in the pathogenesis and aggravation of IBD [188,191]. There are two possible mechanisms for the translocation of oral bacteria into the gut: the haematogenous route and the enteral route. Oral dysbiosis (characterized by an increase in Gram-negative bacteria) can lead to translocation of these bacteria and their virulence factors into the bloodstream. This leads to an increased immune and inflammatory response at the systemic level [9]. The function of the intestinal barrier may also be impaired. Studies in mice have shown that oral administration of *Pg* increased intestinal barrier permeability through down-regulation of tight junction proteins, resulting in significant alteration of the gut microbiome. In addition, mice inoculated with *Pg* exhibited intestinal and systemic inflammation caused by *Pg*-derived endotoxins, such as LPS [192,193]. Another possible route of dissemination of oral bacteria is enteral dissemination. IBD patients have chronic intestinal inflammation with an altered intestinal barrier that leads to increased colonization of oral bacteria at the intestinal level [188,189]. The impact of these oral pathogens in the gut can lead to worsening of chronic gut inflammation. Inoculation of the oral microbiome of IBD children into germ-free mice resulted in enrichment of *Fusobacterium*, *Veillonella* and *Klebsiella* spp. in the gut. Ectopic colonization of these bacteria and in particular *Klebsiella* spp. caused potent T helper 1 (TH1) cell differentiation and inflammation in the gut [194]. Mechanisms of ectopic colonization of the gut by oral bacteria and their role in pathogenesis of IBD still need to be clarified. A better understanding of the oral microbiota in IBD could initiate the development of new diagnostic and therapeutic tools and strategies to target oral bacteria in the management of IBD.

### 3.7. Alzheimer’s Disease

Alzheimer’s disease (AD) is a neurodegenerative disease. It causes progressive and irreversible loss of memory (amnesia), executive function, language (agnosia), and Spatio-temporal orientation disorders leading to dementia and death. It is the most common cause of dementia, with a prevalence of approximately 60–80% worldwide [195,196]. AD is characterized, on the one hand, by the formation of amyloid plaque (also called senile plaque) caused by an extracellular accumulation of the nerve cell-toxic beta-amyloid peptide. On the other hand, neurofibrillary degeneration is related to the accumulation of phosphorylated tubulin-associated unit (TAU) proteins [197]. Inflammation plays an important role in the development and progression of AD. Indeed, there is a close link between beta-amyloid proteins, oxidative stress, and neuroinflammation in the brain, which leads to the loss of neurons and disease progression [198]. The most commonly cited risk factors are genetic and environmental [199]. In recent years, numerous studies have suggested the involvement of oral microbiota in AD [200,201,202]. Patients with AD present a dysbiotic oral microbiota with a decrease of bacterial diversity and an increase of prevalence of the *Moraxella*, *Leptotrichia*, and *Sphaerochaeta* genera [203]. Moreover, serum antibodies to seven oral bacteria (*Aa*, *Campylobacter rectus*, *Fn*, *Pg*, *Pi*, *Tf*, and *Td*) have been found at a higher level in patients with AD, compared with controls [204]. Correlation between the presence or absence of periodontitis and AD has been reported [205,206]. Tooth loss is positively associated with an increased risk of dementia in adults [207]. Two mechanisms may contribute to the progression of AD via the dysbiotic oral microbiota. First, the interaction between oral bacteria or meta factors (e.g., LPS) and the host response leads to a significant production of inflammatory molecules at the systemic level via the TLR-4/NF-κB signaling pathway [208,209]. Systemic inflammation and pro-inflammatory mediators such as CRP, TNF-α, IL-6, and IL-1, are able to activate microglia, which induce increased production of pro-inflammatory molecules and exacerbate neuroinflammation leading to neurodegeneration [210]. The other mechanism by which oral bacteria may contribute to brain inflammation is through bacterial translocation. Not only pro-inflammatory cytokines but also Gram-negative oral bacteria and their virulence factors (LPS) can enter the brain via the blood-brain barrier. LPS from *Pg* and *Td* have been found in post-mortem human brains with AD [211]. Studies on mice have shown the presence of *Pg* in the brain after oral injection. Exposure to *Pg* induced Alzheimer’s disease-like phenotypes in mice with neurodegeneration, amyloid accumulation, impaired cognitive function and reduced learning and memory [210,211,212,213]. Finally, a person with dementia or AD has difficulty maintaining good oral hygiene or regular follow-ups with the dentist, which increases the risk of poor oral health and oral dysbiosis. Also, a recent study exploring the relationships between AD and periodontal treatment, showed that periodontal treatment had a favorable effect on AD-related brain atrophy explored by magnetic resonance imaging [214]. 

### 3.8. Autism Spectrum Disorders

Autism spectrum disorders (ASD) are complex neurodevelopmental disorders that manifest themselves in the first two years of life. They are characterized by persistent deficits in attention, language, and social interaction, often accompanied by restricted interests and repetitive behavior [215]. The prevalence of ASD has been steadily increasing over the past decades. Currently, 1% of the general population is affected by ASD [216]. The causes of autism are still poorly defined. The involvement of several risk factors has been highlighted: genetic, environmental, neuro-pathological, inflammatory, and immunological. Also, maternal prenatal infection and high levels of pro-inflammatory cytokines increase the risk of ASD in children. Concerning microbiota, studies have focused mainly on the relationship between dysbiosis of the intestinal microbiota and ASD, highlighting the microbiota-gut-brain axis in the animal model. However, an increasing number of studies are looking into the role played by the oral cavity and its microbiota. Children with ASD do not have specific oral characteristics related to their pathology, but limited communication, self-neglect, self-mutilation, and eating habit (soft and sweet food) disorders have consequences on their oral health. They have poorer oral hygiene, a higher prevalence of caries, bruxism, and erosions due to frequent regurgitations [217,218,219]. Pain hypersensitivity and communication difficulties lead to delayed oral health care. A recent study showed a difference in the composition of the oral microbiota in ASD patients compared to controls. Patients with ASD showed a decrease in bacterial diversity, an increase in the proportion of pathogenic bacteria such as *Haemophilus* and *Streptococcus*, and a decrease in the proportion of commensal bacteria such as *Alloprevotella*, *Prevotella*, *Selenomonas*, *Actinomyces*, *Porphyromonas*, and *Fusobacterium* [220]. Furthermore, a similar change in the gut microbiota in children with ASD has also been observed, suggesting a potential interaction of the oral microbiota on the gut microbiota [220,221,222]. To date, the direct relationship between ASD and oral microbiota has not been clearly established. However, saliva can be used as a biomarker to help diagnose ASD and allow early management. Recently, the measurement of salivary poly-omic RNA has been described as a new approach to identify children with ASD [223] accurately. Finally, dysbiosis of the microbiota may modulate social behavior, and conversely, restoration of a healthier microbiota may improve ASD symptoms [224,225]. Wang et al. reported that oral probiotics could reduce ASD-like behaviors in offspring induced by maternal immune activation [226].

### 3.9. Oral Mucosal Disease

Oral mucosal disease (OMD) is a series of disorders or conditions affecting the oral soft tissues and mucosa. OMD mainly includes oral infectious diseases such as oral candidiasis, oral lichen planus (OLP) or oral mucosal patches striae diseases, ulcerative lesions such as recurrent aphthous ulcers (RAU), oral premalignant lesions like oral leukoplakia (OLK), and oral cancers, mainly oral squamous cell carcinoma (OSCC) [227]. OMDs share multiple risk factors, including genetic predisposition, immunological disturbances, viral and bacterial infections, food allergies, and nutritional deficiencies, and also hormonal imbalance, mechanical repetitive injuries, and stress. Even though their etiopathogenesis is not yet fully understood, it is well admitted that these different risk factors have the ability to disrupt the diversity and composition of the commensal oral microbiota leading to oral dysbiosis [228].

#### 3.9.1. Oral Microbiota and Oral Lichen Planus

Oral lichen planus (OLP) is a common chronic T-cell-mediated inflammatory oral mucosal disease that occurs in approximately 0.5–2% of the general adult population, with a higher prevalence in women than in men [229]. OLP affects the buccal mucosa, but also the gingiva, tongue, and lips, and in some cases, skin, nails, eyes, and urogenital mucosa. There are essentially two forms of OLP lesions: reticular or non-erosive and erosive. The erosive and non-erosive but atrophic forms of OLP may cause soreness and discomfort and have also been reported as being at risk of malignant transformation in 1–2% of cases [223]. Recent studies on the composition of the oral microbiome focusing particularly on the mycobiome in salivary samples showed that OLP patients (both erosive and reticular forms) had, overall, lower biodiversity of their oral fungal community but an increase in the abundance and frequencies of the genera *Candida* and *Aspergillus* [230]. Interestingly, however, the patients’ bacteriome was significantly more diverse than that of healthy subjects. Also, *Candida* was positively correlated with several anaerobic, periodontopathogen bacterial genera such as *Treponema*, *Bacteroides*, *Aggregatibacter*, *Captnocytophaga*, and *Veillonella* [230]. *Candida*, by its high level of O_2_ consumption permitting the formation of an anaerobic micro niche for these strictly anaerobic bacterial species.

#### 3.9.2. Oral Microbiota and Recurrent Aphthous Ulcers

Recurrent aphthous ulcers (RAU) or recurrent aphthous stomatitis (RAS), characterized by shallow round painful ulcers, are the most common form of oral ulcerative diseases that affect as much as 5–20% of the population [231]. Using new tools to investigate the oral microbiome, such as throughput sequencing and pyrosequencing, has led to the identification of potential bacterial candidates responsible for RAU. The increase in *E. Coli* and *Alloprevotella* associated with a decrease of *Streptococcus* seems to be responsible for RAU [232]. When comparing the oral microbiota of healthy and RAU subjects, the salivary microbiota of RAU subjects tended to contain decreased Firmicutes and increased Proteobacteria. Also, a decrease in *Streptococcus salivarius* and an increase in *Acinetobacter johnsonii* seem to be associated with RAU risk [233]. Finally, when comparing erosive OLP and RAU lesions, studies found that their microbiome was significantly different. The microbiome changes may thus be more related to the diseases themselves rather than to the oral lesions observed clinically. For example, *Streptococcus* and *Sphingomonas* were two of the most abundant bacterial species found in the saliva of OLP patients but were also two of the less abundant species found in the saliva of RAU patients [234]. Taken together, these findings could lead to the development of future diagnostic tools and offer new insight on host immunologic responses and possible cause-effect relationships of these disorders.

#### 3.9.3. Oral Microbiota and Oral Leukoplakia (OLK) and Oral Squamous Cell Carcinoma (OSCC)

Oral leukoplakia (OLK) is defined as a predominantly white lesion of the oral mucosa that cannot be wiped off the mucosa or ascribed to any specific disease process. OLK is often considered a premalignant OMD [228]. Oral squamous cell carcinoma (OSCC) represents the most common cancer of the oral cavity and comprises 80-90% of head and neck cancers with a 50% survival rate at 5 years [235]. The role of specific microbiotas on the development of cancers can be explained through different mechanisms, especially the production of toxins, loss of hormonal homeostasis, and immune tolerance, but also induction of chronic inflammatory signals and carcinogenic metabolites [236]. Several studies on the oral microbiome have been carried out to identify potential microbiota changes and bacteria candidates specific to OLK and OSCC [237,238]. Also, there seems to be a shift in the oral microbiome profile associated with disease progression from OLK to OSCC and during cancer progression from stage 1 to stage 4 [238,239]. Such as in OLP, colonization with *Candida albicans* is common. Studies have shown that, even though bacteria colonization patterns are highly variable, increased abundance of Fusobacteria and reduced levels of Firmicutes are found in OLK [240]. When comparing the salivary microbiome of OLK patients to OSCC subjects, researchers have shown that OSCC subjects have a higher abundance of the phylum Bacteroidetes and *Solobacterium* genus compared to the OLK group, whereas the Streptococcus genus was significantly lower in the OSCC group compared to OLK [241]. This microbiome shift could be a novel precursor marker of malignant transformation of a precancerous lesion [227]. Despite recent data and research progress in the field liking microbiome changes and OMDs, several challenges remain in order to confirm this correlation. Also, confirmation will be essential in order to the make the oral microbiome the target of future diagnostic tools and preventive treatments for OMDs modulation therapies.

## 4. Oral Cavity as a Reflection, Target, and Actor of Our General Health

As summarized in Figure 2, the literature shows increased evidence on the links between the oral cavity, its dysbiotic microbiota, and systemic diseases [8,157,242]. Oral diseases are now considered to be the consequence of a deleterious change in the balance of the oral microbiota making oral microbiota identification and management a major research axis of great interest for practitioners (Figure 3).

### 4.1. Saliva as a Diagnostic Tool

Because it is easily accessible in a non-invasive and inexpensive way, saliva is becoming the new fluid of major interest in the detection of systemic parameters [243]. Salivary biomarkers allow the implementation of novel strategies to help diagnose, limit the prevalence of systemic diseases, and offer new therapeutic tools. The best example came in the past 18 months, with the SARS-CoV-2 infection. Not only is the oral cavity the siege site of the viruses side effects (infection, loss of taste, mucosal damage) and a probably underestimated site for SARS-CoV-2 infection [244], it also offers, with saliva analysis, a quick, reliable, and simple method for collecting samples compared to the nasopharyngeal test [245]. Several kits are already commercialized and used chair side to detect caries (CRT bacteria^®^), periodontal diseases, or IL-6 polymorphism (MyPerioPath^®^, My PerioID^®^) or to determine the risk of developing OSCC (OraRisk^®^ HPV test) [246]. Furthermore, in the field of cancer, the development of techniques such as liquid biopsies and saliva liquid biopsies are being investigated with promising results, especially in lung cancer [247,248,249]. Because saliva is easier to collect, store, and analyze, research in salivaomics will have an important impact in the future to diagnose systemic diseases and potentially monitor disease relapse or aggravation [249,250].

Further technological innovation could reside in the creation of biosensors incorporated in the teeth and measuring the levels of oral biomarkers or quantify enzymes or bacteria present in the oral cavity. These biosensors may be the next approach to monitor an individual’s metabolic status continuously. Mannoor et al. proposed a first version of a “connected tooth” [251]. These authors reported the “grafting” onto a tooth of a graphene “nano sensor” of antimicrobial peptides (antimicrobial peptides or AMP) existing in nature to detect certain species such as *Escherichia coli*, *Helicobacter pylori*, or *Staphylococcus aureus*. They showed that it was possible to detect the binding of a single bacterium on a sensor, confirming the idea that biosensors incorporated in the teeth could possibly be used to monitor disease status constantly. This would support David Wong’s proposition that "oral fluid (saliva), the mirror of the body, is a perfect medium to explore for health and disease surveillance" [252].

### 4.2. Novel Therapeutic Strategies

At the dawn of new, less invasive medicine, one wonders about the benefit of broad-spectrum eradication of oral bacteria and the increase in microbial resistance due to multiple antiseptic and antibiotic treatments [253,254,255,256]. The problem for healthcare practitioners is to offer innovative therapies to resolve the prevalence and co-morbidity of oral and systemic diseases. The search for and development of new therapeutic approaches should, however, not undermine the use of existing therapies such as oral health education, restorative dentistry, and non-surgical periodontal therapies in controlling oral dysbiosis and restoring a healthy environment [257,258]. 

Unfortunately, such “classic” treatments are often insufficient in maintaining oral eubiosis [259]. Thus, new preventive strategies using pre/probiotics are emerging [260]. Numerous studies on the oral microbiota allowed scientists and mathematicians to generate a large database that can lead to the identification of new probiotics from oral microbiota in metabolic regulation. Already, specific probiotics such as *Bifidobacterium pseudocatenulatum* and *Bifidobacterium catenulatum* are used to treat liver damage by attenuating D-galactosamine [261]. Also, in mice, treatment with the probiotic *Bifidobacterium pseudocatenulatum* reduces obesity and inflammation by improving the epithelial barrier of the oral cavity [262,263]. In the oral cavity, guided periodontal pocket recolonization associated with periodontal non-surgical treatment reduced local inflammation and improved long-term clinical outcomes [264]. For example, nitrate used as a prebiotic or a symbiotic with nitrate-reducing probiotics could be promising in preventing oral disease and improving systemic conditions such as hypertension and diabetes, as shown in an in vitro study [265].

Another approach relies on host modulation therapy in order to control and reduce local inflammation, thus reducing systemic inflammation [266,267]. AMPs can affect the homeostasis of the oral cavity through the broad or selective killing of bacteria, but also, due to their immunomodulatory properties, they can influence both the innate and adaptive immune response [268,269]. The most studied AMPs are Leucine leucine-37 (LL-37), α-, and β-defensins. A recent systematic review on the subject reveals that, in saliva, some AMPs such as LL-37, HNP1-3 (Human Neutrophil Peptide 1–3), substance P, adrenomedullin, azurocidin, and some others were increased in periodontal disease, while others like calcitonin gene-related protein or neuropeptide Y were decreased [270]. They could thus be used either as novel disease markers or designed to target specific oral bacteria. Next to AMPs, other fields are investigated, and recently, an oxygen transporter derived from the marine lugworm *Arenicola marina*, HEMARINA-M101 (M101), was tested. M101’s anti-inflammatory and anti-infectious potential, based on its anti-oxidative and tissue oxidation properties, have been tested in vitro on biofilm cultures containing *Pg* and in vivo in a *Pg*-induced subcutaneous calvarial abscess in a mouse model [271]. The results showed that M101 significantly reduced the release of pro-inflammatory cytokines and also had an anti-bacterial effect on *Pg*, confirming its pro-healing properties and making it a potential therapeutic agent in periodontal wound healing and regeneration [271].

Also, oral and chronic systemic diseases share nutrition as a risk factor. Even if the role of micronutrients and vitamins remains to date still unclear, any change in oral health should be considered as a warning sign in the prevention of the development of systemic diseases [272]. For example, in in-vivo animal models, Resolvin E1, an endogenous anti-inflammatory lipid mediator derived from Omega-3 eicosapentaenoic acid, seems to suppress bone loss and restore systemic levels of IL-1β and CRP, also attenuating the inflammatory signal leading to periodontal destruction with no unwanted side-effects [273,274]. Another example is the level of vitamin D and its role in periodontal disease and metabolic disorders [275,276,277,278]. The relationship between vitamin D deficiency and insulin resistance could be explained by inflammation, as vitamin D deficiency is associated with increased inflammatory markers, initial insulin resistance, and subsequent onset of diabetes caused by the death of β cells. In fact, we now know that chronic periodontitis [279] is associated with a low level of vitamin D and, more precisely, with a low serum level of 1.25 (OH) 2D. This type of link corresponds to the associations previously reported between vitamin D and other inflammatory diseases [280,281]. Epidemiological studies have shown an association between a low serum concentration of 25-hydroxyvitamin D3 (25 (OH) D3) and an increased risk of metabolic syndrome and Type 2 diabetes, which may be explained in part by an increase in blood pressure [278]. Importantly, genetic polymorphisms in genes linked to vitamin D can predispose to impaired glycemic control and T2D [282].

Finally, identifying new bacteria candidates using novel techniques such as next-generation sequencing and new delivery systems designed specifically for targeting oral bacteria (i.e., caries or periodontal disease-associated oral dysbiosis) could lead to the development of vaccines in the near future [283,284,285].

## 5. Conclusions

Oral microbiota is a complex organism acquired from birth, influenced by environmental, genetic, and behavioral factors. It interacts with the host’s other microbiotas and is important in health and disease. Diagnosing, controlling, and treating oral dysbiosis can have an impact on systemic diseases and patients’ oral and general wellbeing [286]. It also has an impact on society from a socio-economic point of view [287,288,289].

Data from the literature concerns many medical specialties such as cardiology, angiology, dermatology, and endocrinology, and even if future studies are needed in order to fully understand the link and assess the impact of oral health on general health and vice versa, oral specialists should be aware of its existence and work together with other medical specialists to better serve their patients’ needs and improve their quality of life.

## Figures and Tables

**Figure 1 diagnostics-11-01376-f001:**
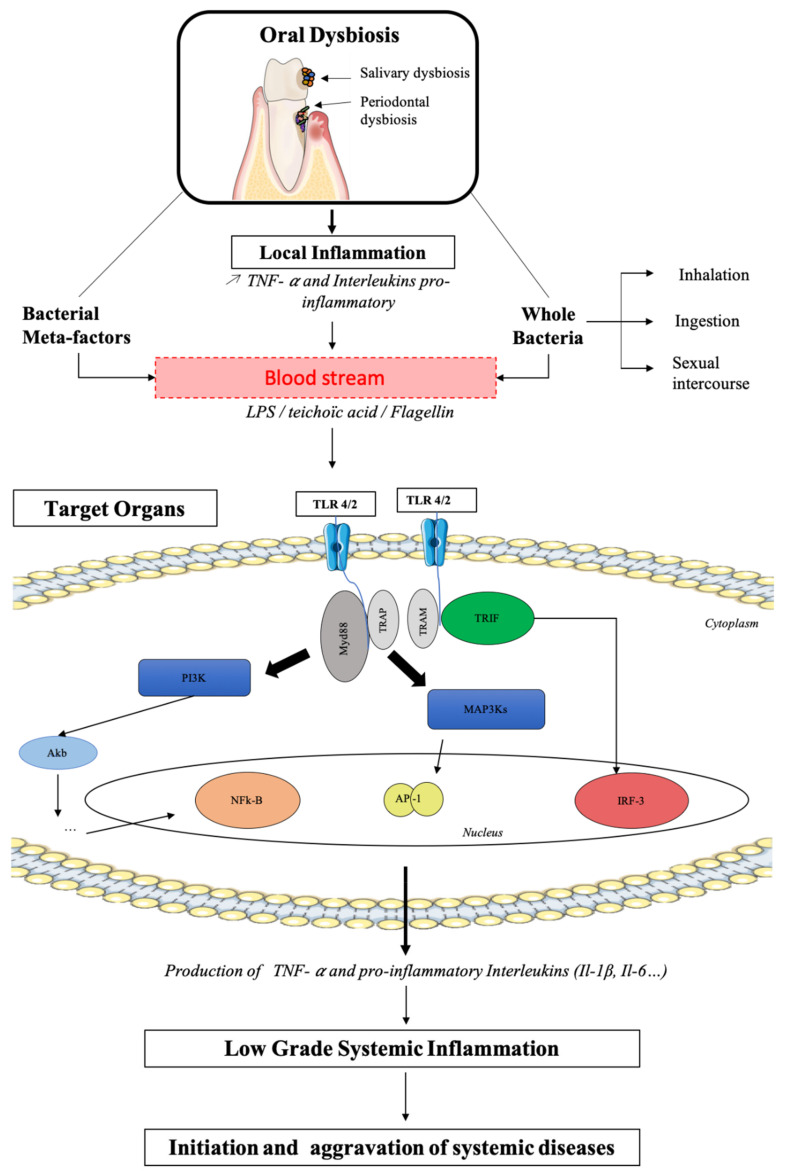
Possible physiopathological mechanisms linking oral dysbiosis to systemic diseases.

**Figure 2 diagnostics-11-01376-f002:**
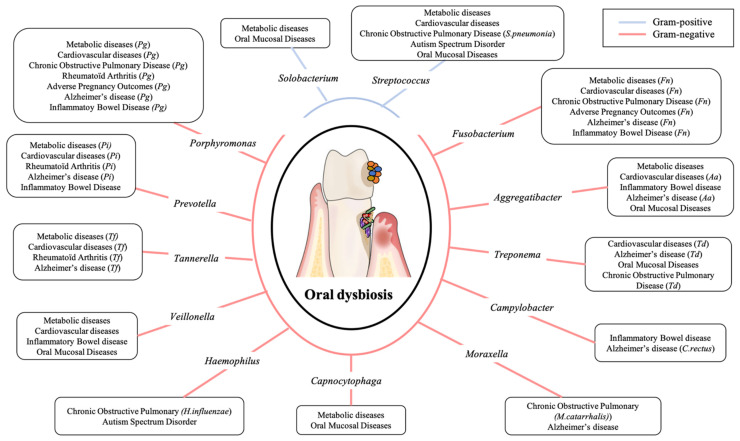
Principal oral pathogens implicated in systemic diseases.

**Figure 3 diagnostics-11-01376-f003:**
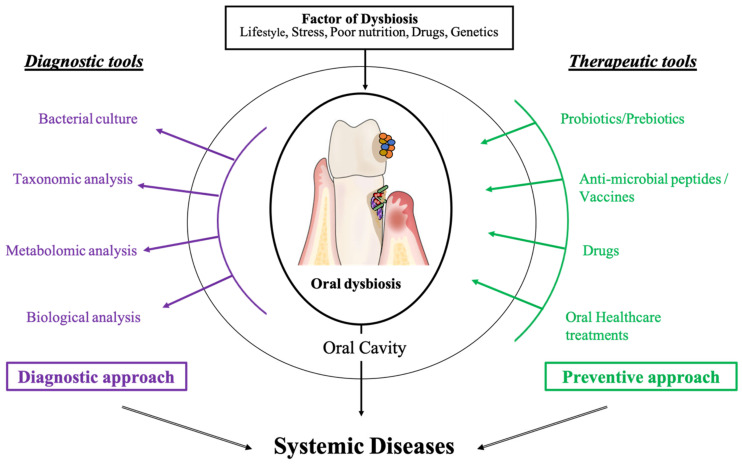
The oral microbiota is a biomarker, diagnostic tool, and target for systemic disease diagnosis and treatment.

## Data Availability

Not applicable.

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
