# Peer review of "Oral Microbiota: A Major Player in the Diagnosis of Systemic Diseases"

_diagnostics, 2021, doi:10.3390/diagnostics11081376_

Round 1

Reviewer 1 Report

Congratulations to the Authors of a great article. They performed a titanic work. I will definitely use it as a source of organized, up-to-date knowledge on the subject discussed by the Authors.

Author Response

Dear Editor,

Thank you for giving us the opportunity to send back a revised version of our manuscript “Oral Microbiota: a Major Player in the Diagnosis of Systemic Diseases”, diagnostics-1320866

We have carefully studied the comments made and have modified the text accordingly as suggested.

Thus, we hope that you will now find our revised version suitable for publication in your journal.

Yours sincerely,

Vincent Blasco-Baque; DMD and PhD

We greatly appreciate the general positive attitude and took into consideration all the suggestions made by the reviewers and Editor to improve our manuscript. We really hope that our answers meet your expectations. All our modifications to the text are underlined to be easily followed.

Reviewer #1:

Congratulations to the Authors of a great article. They performed a titanic work. I will definitely use it as a source of organized, up-to-date knowledge on the subject discussed by the Authors

Thank you for your positive comments

Reviewer 2 Report

The manuscript by Thomas and co-workers aims to highlight the connection between oral specific diseases, oral microbiome dysbiosis and the consequences to an immune response to oral microbes on systemic conditions. They review literature evidence related to several important systemic diseases including diabetes, inflammatory bowel syndrome and Alzheimer disease among others. I found the review interesting and overall easy to follow.

The major concern I have is that the article requires some proof reading. There are several typos and repeats of words in the manuscript. I list some below. In addition, some conceptual mistakes were made which require rewriting of some paragraphs.

line 86 to 89: I think the word mature is not the right word to describe bacteria that enter the stage of biofilm development. Further, it is not widely accepted that bacteria are part of the AEP, but it is discussed in some publications. The authors should describe the AEP as separate developmental step conditioning the tooth surface for subsequent attachment of bacteria as part of the biofilm developmental process.

line 92: Aways write the whole species name when first mention.

line 93: what are advanced bacteria…later colonizer?

line 94: this makes it sound that biofilm maturation always leads to dysbiosis. This is not the case.

line 97: Caries is.

line 98: What is salivary dysbiosis?

line 109: that made me laugh…I think the authors meant microorganisms.

line 112 to 113: Please cite a modern concept for periodontal disease.  (see: PMID: 33902163)

line 137: metafactors? please explain

line 222 to 223: typo, word repeat

line227: normo?

line 278: typo

line 332: Pseudomonas aeruginosa

line 612: typo

Author Response

Dear Editor,

Thank you for giving us the opportunity to send back a revised version of our manuscript “Oral Microbiota: a Major Player in the Diagnosis of Systemic Diseases”, diagnostics-1320866

We have carefully studied the comments made and have modified the text accordingly as suggested.

Thus, we hope that you will now find our revised version suitable for publication in your journal.

Yours sincerely,

Vincent Blasco-Baque; DMD and PhD

We greatly appreciate the general positive attitude and took into consideration all the suggestions made by the reviewers and Editor to improve our manuscript. We really hope that our answers meet your expectations. All our modifications to the text are underlined to be easily followed.

Reviewer #2:

Point 1: line 86 to 89: I think the word mature is not the right word to describe bacteria that enter the stage of biofilm development. Further, it is not widely accepted that bacteria are part of the AEP, but it is discussed in some publications. The authors should describe the AEP as separate developmental step conditioning the tooth surface for subsequent attachment of bacteria as part of the biofilm developmental process.

We thank the reviewer for the comment and we modified the paragraph:

Lines 87 to 92:” … dental plaque is the result of the heterogeneous accumulation of aerobic and anaerobic bacteria that form an adherent deposit on the surface of teeth and oral mucosa [20]. Dental surfaces are covered with an organic film called the Acquired Exogenous Pellicle (AEP) which protects teeth from mechanical and acidic aggressions but which can also condition the tooth surface to promote further attachment of bacteria as part of the biofilm developmental process.”

Point 2: line 94: Always write the whole species name when first mention.

 We thank the reviewer for the comment and we corrected these sentences in the text:

Lines 94 to 97:The first step in the creation of a biofilm is the irreversible adhesion of pioneer bacteria such as Streptococcus gordonii, Streptococcus oralis or Streptococcus mitis to the AEP, which will then proliferate.”

Point 3: line 93: what are advanced bacteria…later colonizer?

We thank the reviewer for the comment and we corrected these sentences in the text:

Lines 97 to 98: “Secondly, these pioneer bacteria will allow the aggregation of new bacteria called late or secondary colonizers [21].

Point 4: line 94: this makes it sound that biofilm maturation always leads to dysbiosis. This is not the case.

We agree to the reviewer’s remark and we corrected these sentences in the text:

Lines 98 to 100: “As the biofilm matures, an equilibrium is created between each bacteria present. Microbial dysbiosis can occur due to local and systemic diseases resulting in a gradual shift toward bacteria with specific profiles.”

Point 5: line 97: Caries is.

 We thank the reviewer for the comment and we corrected these sentences in the text:

Lines 102 to 103: “Caries is responsible for the destruction of the tooth’s hard tissues.”

Point 6: line 98: What is salivary dysbiosis?

We modified the sentence in the text:

Lines 103 to 106:” They are caused by salivary dysbiosis (reduction of the bacterial diversity) resulting from a disturbed supra-gingival biofilm associated with an excess of sugar consumption and/or poor healthcare (factors of dysbiosis).”

Point 7: line 109: that made me laugh…I think the authors meant microorganisms.

We thank the reviewer for the comment and we corrected these sentences in the text:

Lines 115: “In contrast to the rest of the oral cavity, the microorganisms that develop …”

Point 8: line 112 to 113: Please cite a modern concept for periodontal disease.  (see: PMID: 33902163)

We thank the reviewer for this smart remark and added in the text the following sentences in lines 123 to 127:

However, metagenomic sequencing has made it possible to highlight new concepts concerning periodontal dysbiosis associated with periodontitis. This dysbiosis is the result of a qualitative and quantitative modification of a polymicrobial community including commensal and pathogenic bacteria [32].”

Reference:

32.       Scannapieco, F.A.; Dongari‐Bagtzoglou, A. Dysbiosis Revisited. Understanding the Role of the Oral Microbiome in the Pathogenesis of Gingivitis and Periodontitis: A Critical Assessment. J Periodontol 2021, JPER.21-0120, doi:10.1002/JPER.21-0120.

Point 9: line 137: metafactors? please explain

We thank the reviewer for the comment and we added these sentences in the text:

Lines 142 to 145: “Bacterial metafactors are all the bacterial parts that have an activity of either virulence or activation of the immune system during physiology and physiopathology (for example : LPS, Flagellin, teichoic acid …)”

Point 10: line 353: Pseudomonas aeruginosa

We agree to the reviewer’s remark and we corrected these sentences in the text:

Lines 360 to 363: “Studies showed that Haemophilus influenzae, Streptococcus pneumoniae (Sp), Pseudomonas aeruginosa (Pa) …”

Point 11: major concern I have is that the article requires some proof reading. There are several typos and repeats of words in the manuscript.

We agree to the reviewer’s remark and a proof reading was done by different co-authors to correct the different typos in the manuscript. 
